# Learning Subgoal Representations with Slow Dynamics

**Siyuan Li,**\* **Lulu Zheng,**\* **Jianhao Wang, Chongjie Zhang**
Institute for Interdisciplinary Information Sciences
Tsinghua University, Beijing, China
`{sy-li17,zll19,wjh19}@mails.tsinghua.edu.cn`
`chongjie@tsinghua.edu.cn`

## Abstract

In goal-conditioned Hierarchical Reinforcement Learning (HRL), a high-level policy periodically sets subgoals for a low-level policy, and the low-level policy is trained to reach those subgoals. A proper subgoal representation function, which abstracts a state space to a latent subgoal space, is crucial for effective goal-conditioned HRL, since different low-level behaviors are induced by reaching subgoals in the compressed representation space. Observing that the high-level agent operates at an abstract temporal scale, we propose a slowness objective to effectively learn the subgoal representation (i.e., the high-level action space). We provide a theoretical grounding for the slowness objective. That is, selecting slow features as the subgoal space can achieve efficient hierarchical exploration. As a result of better exploration ability, our approach significantly outperforms state-of-the-art HRL and exploration methods on a number of benchmark continuous-control tasks[1][2]. Thanks to the generality of the proposed subgoal representation learning method, empirical results also demonstrate that the learned representation and corresponding low-level policies can be transferred between distinct tasks.

## 1 Introduction

Deep Reinforcement Learning (RL) has demonstrated increasing capabilities in a wide range of domains, including playing games (Mnih et al., 2015; Silver et al., 2016), controlling robots (Schulman et al., 2015; Gu et al., 2017) and navigation in complex environments (Mirowski et al., 2016; Zhu et al., 2017). Solving temporally extended tasks with sparse or deceptive rewards is one of the major challenges for RL. Hierarchical Reinforcement Learning (HRL), which enables control at multiple time scales via a hierarchical structure, provides a promising way to solve those challenging tasks. *Goal-conditioned* methods have long been recognized as an effective paradigm in HRL (Dayan & Hinton, 1993; Schmidhuber & Wahnsiedler, 1993; Nachum et al., 2019). In goal-conditioned HRL, higher-level policies set subgoals for lower-level ones periodically, and lower-level policies are incentivized to reach these selected subgoals. A proper subgoal representation function, abstracting a state space to a latent subgoal space, is crucial for effective goal-conditioned HRL, because the abstract subgoal space, i.e., high-level action space, simplifies the high-level policy learning, and explorative low-level behaviors can be induced by setting different subgoals in this compressed space as well.

Recent works in goal-conditioned HRL have been concentrated on implicitly learning the subgoal representation in an end-to-end manner with hierarchical policies (Vezhnevets et al., 2017; Dilok-thanakul et al., 2019), e.g., using a variational autoencoder (Péré et al., 2018; Nair & Finn, 2019; Nasiriany et al., 2019), directly utilizing the state space (Levy et al., 2019) or a handcrafted space (Nachum et al., 2018) as a subgoal space. Sukhbaatar et al. (2018) proposed to learn subgoal embeddings via self-play, and Ghosh et al. (2018) designed a representation learning objective using an actionable distance metric, but both of the methods need a pretraining process. Near-Optimal

---

\*Denotes equal contribution

[1]Videos available at `https://sites.google.com/view/lesson-iclr`
[2]Find open-source code at `https://github.com/SiyuanLee/LESSON`

Representation (NOR) for HRL (Nachum et al., 2019) learns an abstract space concurrently with hierarchical policies by bounding the sub-optimality. However, the NOR subgoal space could not support efficient exploration in challenging deceptive reward tasks.

In this paper, we develop a novel method, which LEarns the Subgoal representation with SlOw dyNamics (LESSON) along with the hierarchical policies. Subgoal representation in HRL is not only a state space abstraction, but also a form of high-level action abstraction. Since the high-level agent makes decisions at a low temporal resolution, our method extracts features with slow dynamics from observations as the subgoal space to enable temporal coherence. LESSON minimizes feature changes between adjacent low-level timesteps, in order for the learned feature representation to have the slowness property. To capture dynamic features and prevent the collapse of the learned representation space, we also introduce an additional contrastive objective that maximizes feature changes between high-level temporal intervals. We provide a theoretical motivation for the slowness objective. That is, selecting slow features as the subgoal space can achieve the most efficient hierarchical exploration when the subgoal space dimension is low and fixed. We illustrate on a didactic example that our method LESSON accomplishes the most efficient state coverage among all the compared subgoal representation functions. We also compare LESSON with state-of-the-art HRL and exploration methods on complex MuJoCo tasks (Todorov et al., 2012). Experimental results demonstrate that (1) LESSON dramatically outperforms previous algorithms and learns hierarchical policies more efficiently; (2) our learned representation with slow dynamics can provide interpretability for the hierarchical policy; and (3) our subgoal representation and low-level policies can be transferred between different tasks.

## 2 PRELIMINARIES

In reinforcement learning, an agent interacts with an environment modeled as an MDP $M = (S, A, P, R, \gamma)$, where $S$ is a state space, $A$ is an action space. $P : S \times A \times S \rightarrow [0, 1]$ is an unknown dynamics model, which specifies the probability $P(s'|s, a)$ of transitioning to next state $s'$ from current state $s$ by taking action $a$. $R : S \times A \rightarrow \mathbb{R}$ is a reward function, and $\gamma \in [0, 1)$ is a discount factor. We optimize a stochastic policy $\pi(a|s)$, which outputs a distribution over the action space for a given state $s$. The objective is to maximize the expected cumulative discounted reward $\mathbb{E}_\pi[\sum_{t=0}^\infty \gamma^t r_t]$ under policy $\pi$.

## 3 METHOD

In this section, we present the proposed method for LEarning Subgoal representations with SlOw dyNamics (LESSON). First, we describe a two-layered goal-conditioned HRL framework. We then introduce a core component of LESSON, the slowness objective for learning the subgoal representation of HRL. Finally, we summarize the whole learning procedure.

### 3.1 FRAMEWORK

Following previous work (Nachum et al., 2018; 2019), we model a policy $\pi(a|s)$ as a two-level hierarchical policy composed of a high-level policy $\pi_h(g|s)$ and a low-level policy $\pi_l(a|s, g)$. The high-level policy $\pi_h(g|s)$ selects a subgoal $g$ in state $s$ every $c$ timesteps. The subgoal $g$ is in a low dimensional space abstracted by representation function $\phi(s) : S \rightarrow \mathbb{R}^k$. The low-level policy $\pi_l(a|s, g)$ takes the high-level action $g$ as input and interacts with the environment every timestep. Figure 1 depicts the execution process of the hierarchical policy.

LESSON iteratively learns the subgoal representation function $\phi(s)$ with the hierarchical policy. To encourage policy $\pi_l$ to reach the subgoal $g$, we train $\pi_l$ with an intrinsic reward function based on the negative Euclidean distance in the latent space, $r_l(s_t, a_t, s_{t+1}, g) = -||\phi(s_{t+1}) - g||_2$. Policy $\pi_h$ is trained to optimize the expected extrinsic rewards $r_t^{env}$. We use the off-policy algorithm SAC (Haarnoja et al., 2018) as our base RL optimizer. In fact, our framework is compatible with any standard RL algorithm.

Apparently, a proper subgoal representation $\phi(s)$ is critical not only for learning an effective low-level goal-conditioned policy but also for efficiently learning an optimal high-level policy to solve a given task. As the feature dimension $k$ is low, $\phi(s)$ has a compression property, which is necessary

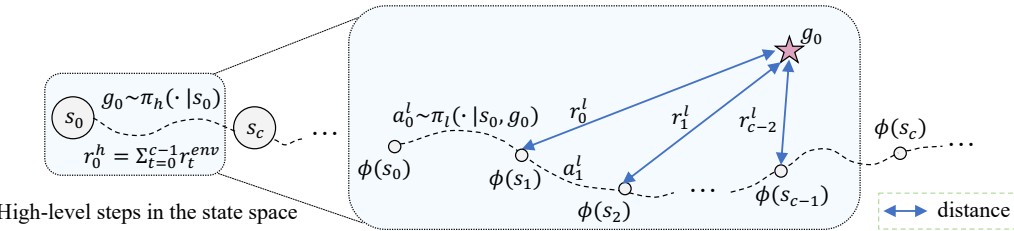

Figure 1: A schematic illustration of the hierarchical policy execution. One high-level step corresponds to $c$ low-level steps. The negative Euclidean distance in the latent space provides rewards for the low-level policy.

to make the hierarchical policy learning easier. If $\phi(s)$ is exactly an identity function without any abstraction, the high-level policy $\pi_h$ still needs to explore in a large space and the complicated subgoal $g$ for the low-level policy is hard to reach as well. In this circumstance, the hierarchical structure cannot simplify the MDP and has no advantage over a flat structure.

## 3.2 Learning Subgoal Representations

Inspired by physics-based priors, features with slow dynamics preserve higher temporal coherence and less noise (Wiskott & Sejnowski, 2002). As the high-level policy acts at a lower temporal resolution compared to the low-level policy, it is sensible to learn a subgoal representation function with a slowness objective. To solve large-scale problems, we parameterize the representation function $\phi(s)$ with a neural network to extract slow features. One natural way of learning $\phi(s)$ is to minimize the squared difference between feature values at times $t$ and $t + 1$,

$$\min_{\phi} \mathbb{E}_{(s_t, s_{t+1}) \sim D}[||\phi(s_t) - \phi(s_{t+1})||_2], \tag{1}$$

where $D$ is a replay buffer. This loss function eliminates fast features, but can be trivially optimized if we allow lossy representation function $\phi$ (e.g., if $\phi(s) = 0$ for $\forall s \in S$). To avoid such trivial solutions and capture dynamic features, we propose a contrastive loss to maximize the distance between high-level state transitions in the latent subgoal space, i.e., $\min_{\phi} \mathbb{E}_{(s_t, s_{t+c}) \sim D}[-||\phi(s_t) - \phi(s_{t+c})||_2]$. To trade off these two loss functions, we adopt the technique of triplet loss (Chopra et al., 2005), i.e., imposing the latent distance between high-level transitions larger than a margin parameter $m$, as shown by Eq. 2. If we remove the margin parameter $m$ and the $max$ operator, Eq. 2 will be dominated by the maximizing distance part. Margin $m$ defines a unit of distance in the latent space, which prevents trivial solutions as well.

$$\min_{\phi} \mathbb{E}_{(s_t, s_{t+1}, s_{t+c}) \sim \mathcal{D}}[||\phi(s_t) - \phi(s_{t+1})||_2 + max(0, m - ||\phi(s_t) - \phi(s_{t+c})||_2)]. \tag{2}$$

The above learning objective abstracts the state space to a latent subgoal space with slow dynamics. As Eq. 2 optimizes the squared difference between feature values, the learned representation can preserve the spatial locality property of the state space, so a subgoal $g$ can be selected in the neighborhood of $\phi(s)$. In the next section, we give a theoretical motivation for the slowness objective. That is, selecting slow features as the subgoal space can promote efficient exploration. Algorithm 1 shows the learning procedure of our method. We update $\phi(s)$ and $\pi_l$ at the same frequency so that the low-level reward function varies in a stationary way. The high-level policy is updated less frequently, as the high-level transitions are less.

## 4 Efficient Exploration with slow subgoal representation

In this section, we provide a theoretical motivation for subgoal representation learning with slow dynamics from a statistical view. To support a formal analysis, we consider selecting a subset of features from the state space as a subgoal space. We prove that, given a fixed subgoal space dimension, selecting slow features as the subgoal space can achieve the most efficient hierarchical exploration. We first define a measure for exploration and describe assumptions of our analysis. Then, we present a theorem about the optimality property and corresponding implications.

---

**Algorithm 1** LESSON algorithm

---

1: **Input:** Number of training steps $N$, margin $m$, replay buffer $D$.
2: **Initialize:** Learnable parameters for $\pi_h(g|s)$, $\pi_l(a|s,g)$ and $\phi(s)$.
3: **for** $t = 1..N$ **do**
4:     Collect experience $(s_t, g_t, a_t, s_{t+1}, r_t^{env})$ under $\pi_h$ and $\pi_l$.
5:     Compute low-level reward $r_t^l = -||\phi(s_{t+1}) - g_t||_2$.
6:     Update the replay buffer $D$.
7:     Optimize $\pi_h$ by maximizing cumulative task rewards with $D$ every $c$ timesteps.
8:     Optimize $\pi_l$ by maximizing cumulative low-level rewards with $D$ every timestep.
9:     Sample a batch of state transitions from $D$ and update $\phi$ with Eq. 2 every timestep.
10: **end for**
11: **Return:** $\pi_h, \pi_l$ and $\phi$.

---

## 4.1 DEFINITIONS AND ASSUMPTIONS

To develop a theoretical analysis, we give a definition of slow features and a measure of exploration. Then, we formulate the exploration process in goal-conditioned HRL as a random walk in the state space as follows.

As our theoretical analysis is broadly applicable to arbitrary feature space, we denote a state $\mathbf{s}_t = [s_t^1, ..., s_t^I]^T$ as a vector containing $I$ features[3]. State $\mathbf{s}_t$ can be factored into slow features $\mathbf{s}_{slow}$ and fast features $\mathbf{s}_{fast}$ with a one-step feature change metric $\Delta s_t^i = |s_t^i - s_{t+1}^i|$ ($1 \leq i \leq I$). Without loss of generality, we assume that $\mathbb{E}_{\pi_r}[\Delta s_t^i] < \mathbb{E}_{\pi_r}[\Delta s_t^{i+1}]$, where $\pi_r$ is a random policy. The expected one-step feature change of slow features is relatively small. With a limited slow feature dimension $k$, $\mathbf{s}_{slow} = [s^1, ..., s^k]^T$, and the rest are fast features. For example, the movements of a robot are slow, but the changes of noisy sensory observations are fast.

**Definition 1** (Measure of Exploration). *In goal-conditioned HRL, an effectiveness measure of hierarchical exploration is defined as the Kullback–Leibler (KL) divergence from the distribution of explored states $q(x)$ to a desired state distribution $p(x)$:*

$$D_{\mathrm{KL}}(p\|q) = \int_{-\infty}^{\infty} p(x) \log\left(\frac{p(x)}{q(x)}\right) dx. \tag{3}$$

In this definition, the desired state distribution $p(x)$ is a prior state distribution while $q(x)$ is the distribution of the states explored by the agent. According to Definition 1, when the state distribution of exploration $q(x)$ is closer to the target state distribution $p(x)$, the exploration is more effective.

**Definition 2** (Random Walk). *In goal-conditioned HRL, the exploration process in the state space is an $I$-dimensional random walk when there is no extrinsic reward for the high-level policy and the low-level policy is optimal. Define $s_0$ as the origin of the state space: $s_0 = \mathbf{0}$, and the unit step of the random walk is $\mathbf{X}_t^c = s_t - s_{t-c}, t = c, 2c, \cdots$, which is i.i.d. Denote a sequence of random variables $\mathbf{Y}_n = \sum_{i=1}^n \mathbf{X}_{ic}^c$, then the asymptotic distribution of $\mathbf{Y}_n$ is $q(x)$: $\mathbf{Y_n} \xrightarrow{D} q(x)$.*

We aim to solve sparse reward problems, where an agent needs to explore with little extrinsic rewards, so we consider the circumstance with no extrinsic rewards and the optimal low-level policy to analyze the exploration problem. Thus the high-level policy selects subgoals randomly. The agent can move independently and identically in the state space, leading to $\mathbf{X}_t^c$ is i.i.d. In fact, $q(x)$ can be seen as the steady state distribution of the Markov chain induced by the policy. To facilitate the analysis of different subgoal representations, we make the following assumptions throughout this section:

(a) The transition function $P(s'|s,a)$ is deterministic.

(b) The features are all independent.

(c) $\mathbf{X}_t^c$ is bounded in the state space: $\{|x_i| \leq r_i, i = 1, \cdots, I\}$, where $x_i$ is the $i$-th element of $\mathbf{X}_t^c$, and $r_i$ is a fixed upper bound of $|x_i|$.

(d) The subgoal $\boldsymbol{g}$ selected by the high-level policy at time $t$ is constrained in the neighbourhood of $s_t$: $\left\{|g_j - s_t^j| \leq r_g, j = 1, \cdots, k\right\}$, where $g_j$ and $s_t^j$ are the $j$-th elements of corresponding vectors, and $r_g$ is a fixed bound of subgoals in all dimensions.

---

[3]The state here refers to a true Markovian state.

Assumption (a) is a general technique to simplify theoretical analysis in RL (Krishnamurthy et al., 2016; Boyan & Moore, 1995). Assumption (b) makes it possible to analyze the exploration of each feature dimension separately. Assumption (c) means that every $c$ timesteps, the agent can move in dimension $i$ with a step size $|x_i| \leq r_i$, and slower features have a smaller bound: $\forall i < i', r_i < r_{i'}$. Taking advantage of the spatial continuity of the state space, subgoals are set in the neighborhood of the current state in the selected subgoal feature dimensions, specified as Assumption (d).

## 4.2 Optimality and Implications

**Theorem 1.** *Assume $p(x)$ is a multivariate Gaussian distribution: $p(x) \sim \mathcal{N}_I(\boldsymbol{x}; \mathbf{0}, \mathbf{R})$, where $\mathbf{R}$ is a diagonal matrix $\mathrm{diag}(r^2)$ and $r$ is large enough. Given a fixed subgoal space dimension $k$, selecting the $k$ slowest features for the subgoal space leads to the optimal hierarchical exploration. Denote the distribution of the explored states in this case as $q_{slow}$, we have:*

$$q_{slow} = q^* = \arg\min_{q \in Q} D_{\mathrm{KL}}(p\|q), \tag{4}$$

*where $Q$ is the sets of all distributions of explored states brought by different subgoal space selection.*

Without any prior knowledge, we assume $p(x)$ is an isotropic Gaussian distribution with zero mean. When $r$ is large enough, $q(x)$ approximates a uniform distribution.

*Proof sketch.* The exploration process is decided by the coverage area of the random walk, as shown in Definition 2, and a larger coverage area leads to better exploration (see Definition 1). We analyze the coverage scale in each dimension separately. The exploration ability varies in different dimensions since the slow-feature dimension has a smaller coverage scale. Notice that the exploration ability in dimension $i$ changes if we select the $i$-th feature for the subgoal space. Concretely, if we choose slow features for the subgoal space, the coverage area in these dimensions will expand. In contrast, selecting fast features decreases the ability of exploration. We prove that with a fixed subgoal space dimension $k$, if and only if we select the $k$ slowest features as the subgoal space, $D_{\mathrm{KL}}(p\|q)$ is minimized, i.e., achieves the optimal hierarchical exploration defined in Definition 1. See the detailed proof in Appendix A.

Selecting slow features as the subgoal space can achieve superior exploration shown in Theorem 1. This property indicates that using the slowness objective to learn the subgoal representation can promote more efficient exploration. As real-world tasks are often on a large scale, utilizing neural networks to extract slow features as the subgoal space is more general. To conclude, Theorem 1 is a theoretical grounding for the learning objective of LESSON.

## 5 Related Work

Learning subgoal representations is a challenging problem in HRL (Dwiel et al., 2019). Nachum et al. (2018) and Zhang et al. (2020) predefined a subspace of observations as a subgoal space with domain knowledge. Li et al. (2019) sought for an alternative way of setting advantage-based auxiliary rewards to the low level policy to avoid this difficult problem. Levy et al. (2019) directly used the whole observation space, which is unscalable to high-dimensional tasks. A variational autoencoder (VAE) (Kingma & Welling, 2013) can compress the high-dimensional observations in an unsupervised way, and it has been utilized to learn a subgoal space in (Péré et al., 2018; Nair & Finn, 2019; Nasiriany et al., 2019). However, the features extracted by VAE can hardly capture the transitional relationship in MDPs. In Vezhnevets et al. (2017) and Dilokthanakul et al. (2019), a subgoal representation is learned in an end-to-end way with hierarchical policies. Since the resulting representation is under-defined, those methods often underperformed (see Nachum et al. (2018)). Ghosh et al. (2018) proposed to learn representations using an actionable distance metric, assuming that goal-conditioned policies are given. Sukhbaatar et al. (2018) developed a method called HSP to learn subgoal representations via self-play, but HSP requires a pretraining process, and thus it may be inefficient. Near-Optimal Representation (NOR) for HRL (Nachum et al., 2019) outperforms the previous methods by learning representations bounding the sub-optimality of hierarchical policies. However, the optimization of NOR is complicated, and the abstraction of the NOR space does not aim for efficient exploration. In contrast, we develop a simple subgoal space learning method with a slowness objective. Furthermore, we formally show that the slowness objective has a theoretical grounding for better exploration ability.

Slowness or temporal coherence has been an important prior for learning state representations in continuous control tasks (Bengio et al., 2013; Jonschkowski & Brock, 2015; Lesort et al., 2018). Standard Slow Feature Analysis (SFA) methods learn slow features by solving an optimization problem with constraints (Wiskott, 1999; Wiskott & Sejnowski, 2002). However, their expressivity tends to scale unfavorably in high-dimensional problems. To increase the expressivity, hierarchical SFA (Franzius et al., 2007; 2011; Escalante-B & Wiskott, 2013) composes multiple SFA modules in a layer-wise way. More recent works use neural networks to extract slow features using a slowness loss function. To avoid trivial solutions, another term, such as reconstruction loss (Goroshin et al., 2015a; Finn et al., 2016) or prediction error (Goroshin et al., 2015b), is also included in the loss function. In similarity metric learning, contrastive or triplet loss is investigated to capture slow features in video and audio datasets as well (Jayaraman & Grauman, 2016; Jansen et al., 2018). In reinforcement learning, several approaches exploit the slowly changing bias to extract useful features so that policy learning can be accelerated (Zhang et al., 2009; Legenstein et al., 2010; Oord et al., 2018). To the best of our knowledge, we are the first to utilize the slowness objective in HRL, and our proposed method significantly outperforms state-of-the-art HRL methods on benchmark environments.

The inductive bias of slowness has largely been investigated in the skill discovery methods as well. Continual Curiosity driven Skill Acquisition (CCSA) learns a latent space with SFA, and utilizes curiosity-driven rewards in this latent space to train skills (Kompella et al., 2017). Similarly, Machado et al. (2017a), Jinnai et al. (2019) and Bar et al. (2020) proposed to learn options to reach local maxima or minima of the Proto-value functions (PVFs) (Mahadevan & Maggioni, 2007). As pointed out by Sprekeler (2011), the objective functions of SFA and PVFs are equivalent, when the adjacent function of PVFs is the transition function in MDP. But obtaining the full transition function in large scale tasks is nearly infeasible. To solve large scale problems, Machado et al. (2017b) and Ramesh et al. (2019) proposed to replace PVFs with eigenvectors of the deep successor representation (Kulkarni et al., 2016), which equal to scaled PVFs. Jinnai et al. (2020) approximated the computation of PVFs with the objective introduced by Wu et al. (2018). Our method and those skill discovery methods share some similarities in learning low-level policies in a smooth or slow latent space. However, the skill discovery methods can be regarded as bottom-up HRL, where a set of task-agnostic low-level skills are firstly learned with some intrinsic reward functions and then composed to solve downstream tasks. In contrast, our goal-conditioned method can be regarded as top-down HRL, where the high-level policy sets subgoals to the low level during learning a task, and the level-level policy is incentivized to reach those subgoals.

# 6 EXPERIMENTS

We conduct experiments to compare our approach to existing state-of-the-art methods in HRL and in efficient exploration. Firstly, we show on a didactic example that LESSON can achieve the most efficient state coverage among all the compared subgoal representations. To demonstrate our strengths in high-dimensional tasks, we then compare with several baselines on a number of benchmark continuous-control tasks. After that, we analyze the dynamic property of the learned subgoal representation and provide an interpretation by visualization. Lastly, we show that both the subgoal representation and the low-level policies learned by our method are transferable.

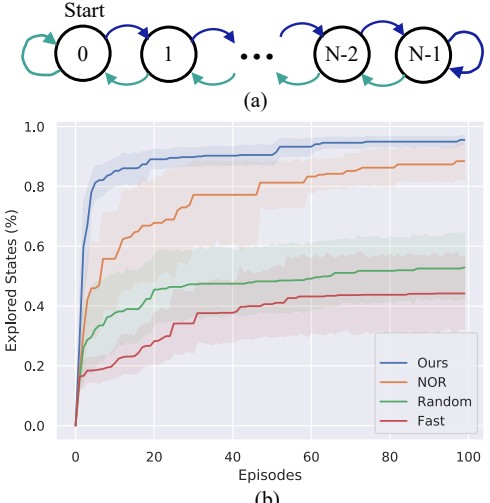

Figure 2: (a) The NChain environment. (b) Results on the 64-link chain environment. Each line is the mean of 20 runs with shaded regions corresponding confidential intervals of 95%.

## 6.1 DIDACTIC EXAMPLE: NCHAIN

The NChain environment was designed hard to explore by Osband et al. (2016), as shown in Figure 2(a). Starting from state 0, an agent can move forward (blue arrow) to the next state in the chain or

backward (green arrow) to the previous state. The state representation is encoded in binary, so the low bits are features with fast dynamics. To make the problem harder, the effect of each action is randomly swapped with a probability of $0.1$. In this near-deterministic environment, we compare the exploration ability of our method to HRL methods using other subgoal spaces with a dimension $k = 1$. Baselines include the NOR subgoal space (Nachum et al., 2019), a randomly selected bit of the state representation and the lowest bit (fast features). For pure exploration comparison, we consider the circumstance of no external rewards, the same as the setting of our theoretical analysis in Section 4. Figure 2(b) illustrates that using slow features as the subgoal space can achieve the most efficient exploration with goal-conditioned HRL. As expected, the performance of fast features as the subgoal space is the worst. Randomly selected features perform better than fast features. Although NOR aims to bound the sub-optimality of the value function, our method outperforms NOR in terms of exploration.

## 6.2 MUJOCO TASKS

We evaluate our proposed subgoal representation learning objective on a set of challenging MuJoCo tasks that require a combination of locomotion and object manipulation. The details of our full implementation and environments are available in Appendix B. We conduct experiments comparing to the following methods in hierarchical learning and exploration, and all the learning curves in this section are averaged over 10 runs.

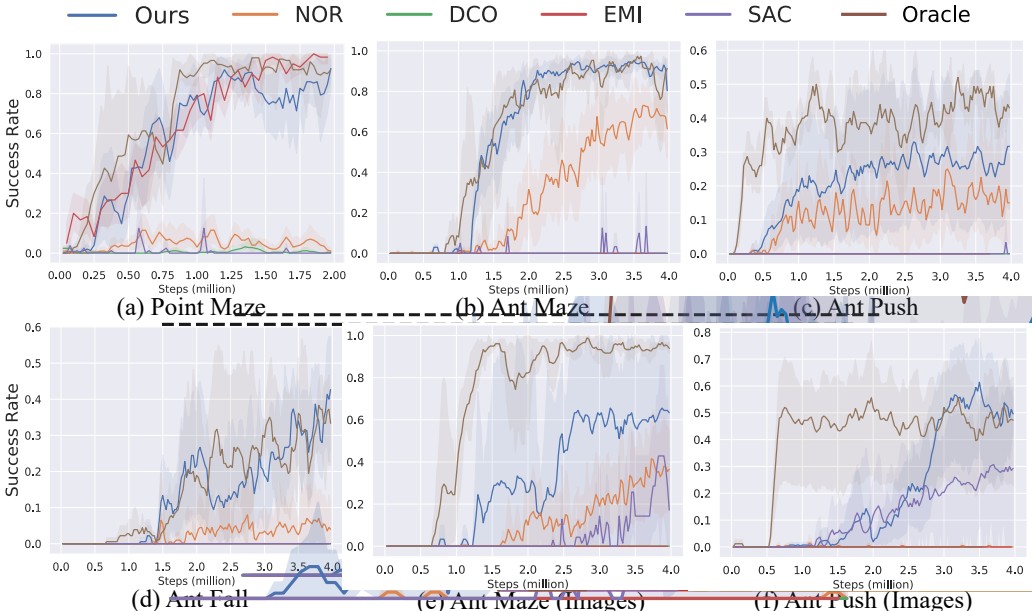

Figure 3: Performance of each method on a suite of MuJoCo environments.

- *NOR*: HRL with a learned subgoal space, which is optimized to bound the sub-optimality of the hierarchical policy (Nachum et al., 2019).
- *Oracle*: HRL with the oracle subgoal space ($x$, $y$ position of the agent) in navigation tasks.
- *DCO*: A hierarchical exploration method with deep covering options (Jinnai et al., 2020)[4].
- *EMI*: A flat exploration method by predicting dynamics in a latent space (Kim et al., 2019).
- *SAC*: The base RL algorithm used in our method (Haarnoja et al., 2018).

Benefiting from a better exploration ability, our method with a temporally-coherent subgoal space significantly outperforms baseline methods in terms of speed and quality of convergence. Even when the raw observation is given by using top-down images, our method can achieve high success rate, presented in Figure 3(e), (f). In the Ant Maze task, our method reaches a success rate of $100\%$ at only $1.5$ million training steps, which is more than two times faster than the NOR algorithm.

---

[4]For a fair comparison, we use the online version of DCO, as the offline version needs a pretraining process.

In the Point Maze task, the flat exploration method EMI shows an equal performance with our approach. However, when the dynamic model is more complex (e.g., for the Ant robot), predicting dynamics becomes much harder, and the performance of EMI degrades dramatically. We evaluate NOR with its published code[5], and results show its ineffectiveness of exploration in challenging tasks. The online DCO method can hardly learn successful policies in those tasks, partly because the pretraining of the second eigenfunction in their method is necessary.

### 6.3 ANALYSIS OF LEARNED REPRESENTATIONS

We visualize the subgoal representation and learned hierarchical policies of our method in the Ant Push task in Figure 4. The learned subgoal space highly resembles the oracle $(x, y)$ position space. By setting subgoals in the learned latent space, the high-level policy guides the agent to jump out of the local optimum of moving towards the goal. The Ant robot under the hierarchical policy firstly moves to the left, then pushes the block to the right, and finally reaches the goal. In contrast, the SAC agent without a hierarchical structure easily gets stuck into the local optimum of moving directly to the goal, since the immediate extrinsic reward is given as the negative L2 distance to the environment goal.

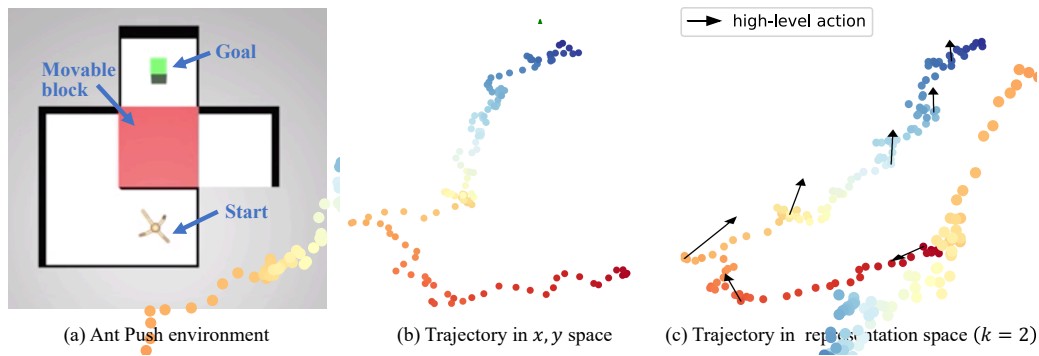

(a) Ant Push environment     (b) Trajectory in $x, y$ space     (c) Trajectory in representation space ($k = 2$)

Figure 4: The color gradient of the trajectory is based on episode timestep (red for the beginning of an episode, blue for the end). Black arrows denoting high-level actions point to the subgoals from the decision-making states. If the Ant robot moves directly towards the goal, it will fail to reach it, as it will push the movable block into the path to the goal.

### 6.4 PARALLEL LEARNING OF THE REPRESENTATION FUNCTION AND POLICIES

We show the subgoal representation learning process in the Ant Push (Images) task in this section. Figure 5 (a)∼(h) visualize trajectories to a hard goal in the representation spaces and the visited areas in the $x, y$ space at different learning stages. Along with the representation visualization, we evaluate an easy goal as the midpoint of the trajectory to the hard goal.

The learning of the hierarchical policy and the subgoal representation could promote each other. At about $0.2$ million timesteps, with the distance-to-goal dense rewards, our method approximately learns an inaccurate subgoal representation and the policy to reach the easy goal. Since the learned representation is generalizable to the neighborhood of the explored areas to some extent, which facilitates the exploration of the hierarchical policy, the explored areas are expanded little by little. The newly collected samples in the expanded region could be utilized to improve the subgoal representation further.

### 6.5 TRANSFERABILITY OF REPRESENTATIONS

Because of the generality of our representation learning objective with slow dynamics, the learned subgoal space is transferable between different tasks of the same robot. The low-level policy induced by the learned subgoal representation is transferable as well. To verify this transferability, we initialize the representation network and the low-level policy network in a target task with those

---
[5]Code at `https://github.com/tensorflow/models/tree/master/research/effici ent-hrl`

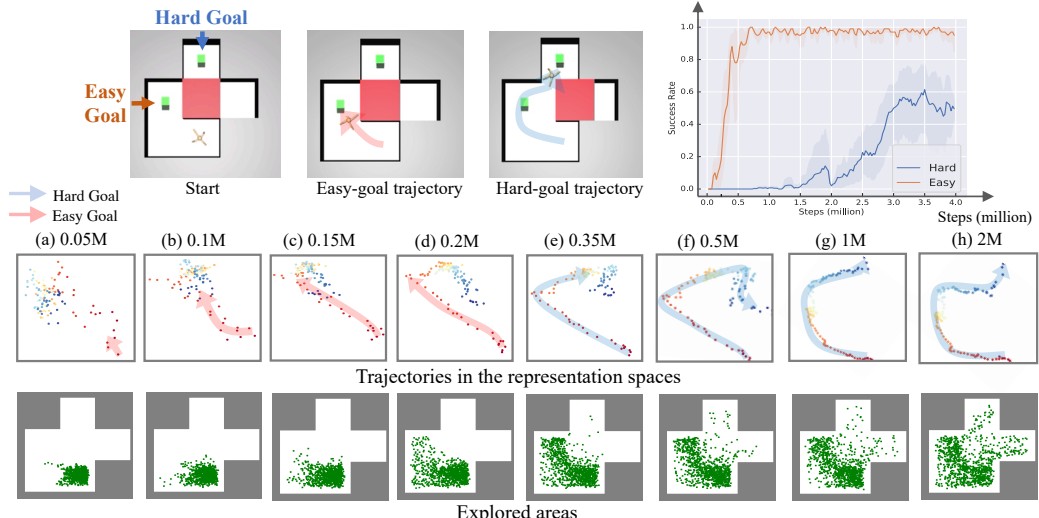

Figure 5: Subgoal representations at different learning stages in the Ant Push (Images) task. The red transparent arrows denote the trajectories from the start to the midpoint (easy goal), while the blue ones denote the trajectories to the hard goal. The explored areas are visualized by 1000 experiences sampled from the replay buffer.

weights learned in a source task and further finetune them in the target task. The high-level policy for the target task is randomly initialized. From Figure 6, we can see that transfer learning helps the agent learn more efficiently and achieve better asymptotic performance.

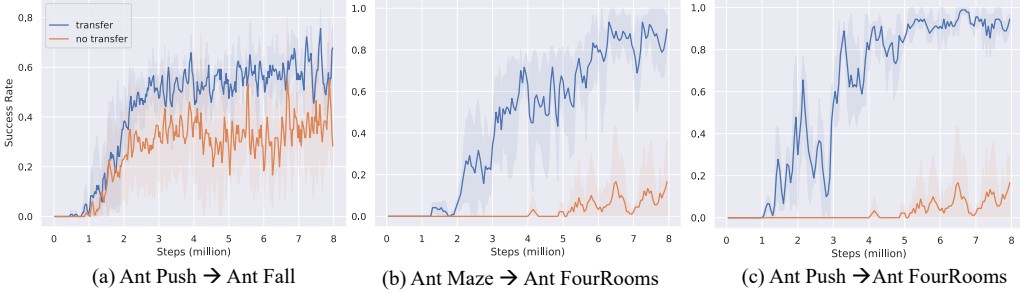

Figure 6: Ant FourRooms is a navigation task in a four-room maze. The source models are randomly picked from the runs shown in Figure 3.

# 7 CONCLUSION

In this work, we propose a self-supervised subgoal representation learning method, LESSON. Our approach is motivated by the slowness prior and supports iterative learning of the representation function and hierarchical policies. In addition, we provide a theoretical grounding for the slowness prior in hierarchical exploration. We test our method on a suite of high-dimensional, continuous control tasks, and it significantly outperforms state-of-the-art HRL and exploration methods. Furthermore, the subgoal representation and low-level policies learned by LESSON are transferable between different tasks. Since the low-level policy learning may result in a non-stationary high-level transition function, combining LESSON with off-policy correction methods to reduce the variance of off-policy learning might be a promising future direction. Furthermore, as the rewards for the continuous control tasks are deceptive and dense, another challenging problem is learning a good subgoal representation and hierarchical policies with extremely sparse rewards.

## ACKNOWLEDGEMENTS

The authors would like to thank the anonymous reviewers for their valuable comments and helpful suggestions. This work is supported in part by Science and Technology Innovation 2030 – "New Generation Artificial Intelligence" Major Project (No. 2018AAA0100904), and a grant from the Institute of Guo Qiang, Tsinghua University.

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

## A  PROOF

**Theorem 1.** *Assume $p(x)$ is a multivariate Gaussian distribution: $p(x) \sim \mathcal{N}_I(\boldsymbol{x}; \mathbf{0}, \mathbf{R})$, where $\mathbf{R}$ is a diagonal matrix $\mathrm{diag}(r^2)$ and $r$ is large enough. Given a fixed subgoal space dimension $k$, selecting the $k$ slowest features for the subgoal space leads to the optimal hierarchical exploration. Denote the distribution of the explored states in this case as $q_{slow}$, we have:*

$$q_{slow} = q^* = \arg\min_{q \in Q} D_{\mathrm{KL}}(p\|q), \tag{4}$$

*where $Q$ is the sets of all distributions of explored states brought by different subgoal space selection.*

First, we prove that $q(x)$ is a multivariate Gaussian distribution regardless of the distribution of $\mathbf{X}_t^c$. Since there is no extrinsic reward, the high-level policy will set subgoals to the low-level randomly every $c$ steps, thus $\mathbf{X}_c^c, \mathbf{X}_{2c}^c, \ldots, \mathbf{X}_{nc}^c$ are independent and identically distributed with the same mean vector $\boldsymbol{\mu} = \mathbb{E}\left[\mathbf{X}_{ic}^c\right] \in \mathbb{R}^I$ and the same covariance matrix $\boldsymbol{\Sigma}_{I \times I}$. Denote the average of $Y_n$ as

$$\frac{1}{n}\mathbf{Y}_n = \frac{1}{n}\sum_{i=1}^{n} \mathbf{X}_{ic}^c = \overline{\mathbf{X}}_n. \tag{5}$$

By *Multidimensional Central limit theorem* (Van der Vaart, 2000), we have

$$\sqrt{n}\left(\overline{\mathbf{X}}_n - \boldsymbol{\mu}\right) \xrightarrow{D} \mathcal{N}_I(\boldsymbol{x}; \mathbf{0}, \boldsymbol{\Sigma}). \tag{6}$$

Plug Eq. 5 into Eq. 6 and consider finite samples, we have

$$\mathbf{Y_n} \xrightarrow{D} \mathcal{N}_I\left(\boldsymbol{x}; n\boldsymbol{\mu}, n\boldsymbol{\Sigma}\right). \tag{7}$$

When $n \to \infty$, the distribution of $Y_n$ converges to $q(x)$, which means $q(x)$ is a multivariate Gaussian distribution. Without loss of generality, we consider the case when $n = 1$, i.e., $\mathcal{N}_I\left(\boldsymbol{x}; \boldsymbol{\mu}, \boldsymbol{\Sigma}\right)$, to compare different KL divergence induced by different subgoal representations. The exploration process can be formulated as a random walk in the state space with continuous action space (Definition 2). The selection of the features for the subgoal space only changes the variance of the unit action $\mathbf{X}_t^c$ in the random walk, furthermore, deciding the covariance matrix of $q(x)$.

Next, we analyze the statistical characteristics of $\mathbf{X}_t^c$. Since all features are independent, the joint distribution is the product of all the marginal distributions: $f_{\mathbf{X}_t^c}(x) = \Pi_{i=1}^{I} f_i(x)$, where $f_{\mathbf{X}_t^c}(x)$ is the Probability density function (PDF) of $\mathbf{X}_t^c$ and $f_i(x)$ is the marginal PDF of $\mathbf{X}_t^c$ in dimension $i$. As Assumption (c) indicates that $x_i \in [-r_i, r_i]$, if not selecting the $i$-th feature for the subgoal space, $f_i(x)$ is a continuous uniform distribution $U[-r_i, r_i]$, so the variance in dimension $i$ can be denoted as $\sigma_i^2 = \frac{r_i^2}{3}$.

However, if we select the $i$-th feature for the subgoal space, the distribution is modified since the low-level policy is optimal (i.e., the agent moves to the subgoal as close as possible during $c$ steps). Besides, notice that by Assumption (d), the $i$-th element of subgoal $g_i$ is uniformly distributed in $[-r_g, r_g]$. Therefore, when $g_i$ lies within the interval $[-r_i, r_i]$, the agent can reach the subgoal within $c$ steps. When $g_i > r_i$ ($g_i < -r_i$), the agent can only reach as far as $r_i$ ($-r_i$). Denote the changed Cumulative Distribution Function (CDF) of $\mathbf{X}_t^c$ in dimension $i$ as $F_i'(x)$, if $r_g > r_i$,

$$F_i'(x) = \begin{cases} 0 & x < -r_i \\ \frac{r_g - r_i}{2r_g} + \frac{1}{2r_g}(x + r_i) & -r_i \le x < r_i \\ 1 & r_i \le x \end{cases}. \tag{8}$$

In contrast, if $r_g \le r_i$, the agent can reach any subgoal within the interval $[-r_g, r_g]$, so we have

$$F_i'(x) = \begin{cases} 0 & x < -r_g \\ \frac{x + r_g}{2r_g} & -r_g \le x \le r_g \\ 1 & r_g < x \end{cases}. \tag{9}$$

In both cases, the mean vector $\boldsymbol{\mu}$ is still $\mathbf{0}$, but the variance $\sigma_i^2$ will increase to $r_i^2 - \frac{2r_i^3}{3r_g}$ when $r_g > r_i$ and $\sigma_i^2$ will decrease to $\frac{r_g^2}{3}$ when $r_g \le r_i$. Denote the selection operation as an operator $\mathbb{S}$, and we

have

$$\sigma_i^2 = \frac{r_i^2}{3},$$

$$\mathbb{S}(\sigma_i^2) = \begin{cases} r_i^2 - \frac{2r_i^3}{3r_g} & r_i \leq r_g \\ \frac{r_g^2}{3} & r_i > r_g \end{cases}. \tag{10}$$

Finally, we want to prove if and only if $q(x) = q_{slow}(x)$, $D_{KL}(p\|q)$ can reach the minimum with the constraint of fixed subgoal space dimension $k$. Consider a distribution $q(x)$ brought by randomly selecting $k$ features from the state space as the subgoal space, since $p(x)$ and $q(x)$ are both multivariate Gaussian distribution (Assumption (b)), the KL divergence from $q(x)$ to $p(x)$ is

$$D_{KL}(p\|q) = \frac{1}{2}\left[\log\frac{\det(\Sigma_q)}{\det(\Sigma_p)} - I + \text{tr}\left(\Sigma_q^{-1}\Sigma_p\right) + (\mu_q - \mu_p)^T \Sigma_q^{-1}(\mu_q - \mu_p)\right], \tag{11}$$

where $I$ is the dimension of $q(x)$, and $tr$ stands for the trace of the matrix (Duchi, 2007). Now we want to prove the KL divergence reaches the minimum if and only if $q(x) = q_{slow}(x)$. Since the covariance matrix $\Sigma$ is symmetric positive definite, there exists a full rank orthogonal matrix $U$ containing of the eigenvectors of $\Sigma$ as its columns and a diagonal matrix $\Lambda$ such that $\Sigma = U\Lambda U^T$ (Horn & Johnson, 2012). Then $\mathcal{N}_I$ can be transformed into a standard multivariate Gaussian distribution through rotation and stretching.

$$Z = B^{-1}(Y - \mu), Z \sim \mathcal{N}_I(0, I), \tag{12}$$

where $Y \sim \mathcal{N}_I(x; \mu, \Sigma)$, $B = U\Lambda^{1/2}$, and $\Lambda^{1/2}$ is a diagonal matrix whose entries are the square roots of the corresponding entries from $\Lambda$ (Do, 2008). Since $q(x)$ is symmetrical, thus rotation won't change the KL divergence, so we only need to consider the case where $\Sigma$ is a diagonal matrix, which can be denoted as below.

$$\Sigma = \begin{bmatrix} \sigma_1^2 & 0 & 0 & \cdots & 0 \\ 0 & \sigma_2^2 & 0 & \cdots & 0 \\ 0 & 0 & \sigma_3^2 & \cdots & 0 \\ \vdots & \vdots & \vdots & \ddots & \vdots \\ 0 & 0 & 0 & \cdots & \sigma_I^2 \end{bmatrix}, \tag{13}$$

where $\sigma_i^2$ is the variance in dimension $i$. Notice $\mu_i = 0$, thus Eq. 11 can be rewritten as

$$\begin{aligned} D_{KL}(p\|q) &= \frac{1}{2}\left[\log\frac{\sigma_1^2\sigma_2^2\ldots\sigma_I^2}{r^{2I}} - I + \sum_{i=1}^{I}\frac{r^2}{\sigma_i^2} + \sum_{i=1}^{I}\frac{\mu_i^2}{r^2}\right] \\ &= \frac{1}{2}\left[\sum_{i=1}^{I}\log\frac{\sigma_i^2}{r^2} - I + \sum_{i=1}^{I}\frac{r^2}{\sigma_i^2}\right] \\ &= \frac{1}{2}\left[\sum_{i=1}^{I}\left(\log\frac{\sigma_i^2}{r^2} + \frac{r^2}{\sigma_i^2}\right) - I\right]. \end{aligned} \tag{14}$$

Consider a function:

$$f(\sigma_i^2) = \log\frac{\sigma_i^2}{r^2} + \frac{r^2}{\sigma_i^2}. \tag{15}$$

It's easy to find that $f(\sigma_i^2)$ is monotonically decreasing when $\sigma_i^2 \in (0, r^2)$ (since $r$ is large enough, the condition is easily met), which means increasing the variance in dimension $i$ will decrease $D_{KL}(p\|q)$. Consider feature dimension $i$ and feature dimension $j$, and $r_i \leq r_j$. Then we prove

$$f(\mathbb{S}(\sigma_i^2)) + f(\sigma_j^2) \leq f(\sigma_i^2) + f(\mathbb{S}(\sigma_j^2)), \tag{16}$$

where $\sigma_i^2 = \frac{r_i^2}{3}, \sigma_j^2 = \frac{r_j^2}{3}$. We consider three cases:

(1) $r_i \leq r_g \leq r_j$. Recall that selecting the $i$-th feature for the subgoal space can increase $\sigma_i^2$ when $r_g > r_i$ while decrease $\sigma_i^2$ when $r_g < r_i$. Therefore, we have:

$$f(\mathbb{S}(\sigma_i^2)) + f(\sigma_j^2) \leq f(\sigma_i^2) + f(\sigma_j^2) \leq f(\sigma_i^2) + f(\mathbb{S}(\sigma_j^2)). \tag{17}$$

(2) $r_g \leq r_i \leq r_j$. Since $f(\mathbb{S}(\sigma_i^2)) + f(\sigma_j^2) = f(\frac{r_g^2}{3}) + f(\sigma_j^2)$, and $f(\sigma_i^2) + f(\mathbb{S}(\sigma_j^2)) = f(\sigma_i^2) + f(\frac{r_g^2}{3})$. Notice that $f(\sigma_j^2) < f(\sigma_i^2)$, then Eq. 16 holds.

(3) $r_i \leq r_j \leq r_g$. Rewrite Eq. 16 as

$$f(\mathbb{S}(\sigma_i^2)) - f(\sigma_i^2) \leq f(\mathbb{S}(\sigma_j^2)) - f(\sigma_j^2)$$

$$\Longrightarrow f\left(r_i^2 - \frac{2r_i^3}{3r_g}\right) - f\left(\sigma_i^2\right) \leq f\left(r_j^2 - \frac{2r_j^3}{3r_g}\right) - f(\sigma_j^2) \tag{18}$$

$$\Longrightarrow \log\left(3 - \frac{2r_i}{r_g}\right) + \frac{6r^2 r_i - 6r^2 r_g}{r_i^2(3r_g - 2r_i)} \leq \log\left(3 - \frac{2r_j}{r_g}\right) + \frac{6r^2 r_j - 6r^2 r_g}{r_j^2(3r_g - 2r_j)}.$$

To prove Eq. 18, we consider another function:

$$g(x) = \log(3 - 2x) + \frac{6t^2 x - 6t^2}{x^2(3 - 2x)}, \tag{19}$$

where $t = \frac{r}{r_g} >> 1$. It's easy to find that $g'(x) > 0$ when $x \in (0, 1)$, which means $g(x)$ is monotonically increasing when $x \in (0, 1)$. Therefore, Eq. 18 holds.

Now we have proved that Eq. 16 holds under all possible conditions, which means selecting the slower features as the subgoal space will lead to a smaller KL divergence, i.e., better exploration. Thus Theorem 1 follows immediately.

## B  EXPERIMENTAL DETAILS

### B.1  ENVIRONMENTS

The environments of Point Maze, Ant Maze, Ant Push, and Ant Fall are as described in Nachum et al. (2019), shown in Figure 7. In each navigation task, we create an environment composed of $4 \times 4 \times 4$ blocks, some movable and some with fixed position. During training, the target locations $(x, y)$ are randomly selected by the environment from all possible points. Final results are evaluated on a single challenging goal denoted by a small green block. For the 'Images' versions of these environments, we zero-out the $x$, $y$ coordinates in the observation and append a low-resolution $5 \times 5 \times 3$ top-down view of the environment, equal to that used in Nachum et al. (2019).

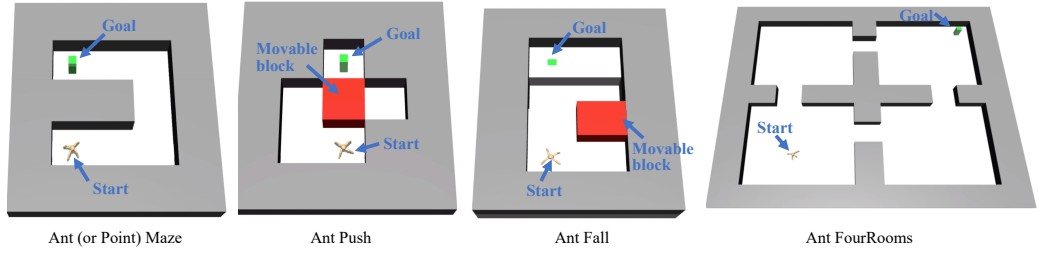

Figure 7: A collection of environments that we use.

The Ant FourRooms task has a much larger maze structure, which is four times as large as the Ant Maze task. So the maximal episode length for Ant FourRooms is also larger, which equals 1000. The maximal episode lengths of the other tasks are 500.

### B.2  NETWORK STRUCTURE

The actor network for each level is a Multi-Layer Perceptron (MLP) with two hidden layers of dimension 256 using ReLU activations. The critic network structure for each level is identical to that of the actor network. We scale the outputs of the actor networks of both levels to the range of corresponding action space with tanh nonlinearities. The representation function $\phi(s)$ is parameterized by an MLP with one hidden layer of dimension 100 using ReLU activations.

### B.3 TRAINING PARAMETERS

- Discount factor $\gamma = 0.99$ for both levels.
- Adam optimizer; learning rate $0.0002$.
- Soft update targets $\tau = 0.005$ for both levels.
- Replay buffer of size $1e6$ for both levels.
- Reward scaling of $0.1$ for both levels.
- Entropy coefficient of SAC $\alpha = 0.2$ for both levels.
- Low-level policy length $c = 10$ for the Point robot and $c = 20$ for the Ant robot except for the Ant Push task. In the Ant Push task, $c = 50$.
- Subgoal dimension of size 2. We train the high-level policy to output actions in $[-10, 10]^2$ when $c = 10$ or $c = 20$ ($[-20, 20]^2$ when $c = 50$). These actions correspond to desired deltas in state representation.

We did not perform a grid search on hyper-parameters, therefore better performances might be possible for these experiments.

### B.4 EVALUATION

Learned hierarchical policies are evaluated every 25000 timesteps by averaging performance over 10 random episodes.

## C ADDITIONAL EXPERIMENTAL RESULTS

Table 1 demonstrates that the dynamics of the features learned by our method are slow. NOR has a relatively good performance in Ant Maze and Ant Maze (Images), since the NOR features in these two tasks are slower than those in other tasks. The state space of the Point robot is low-dimensional and contains little information other than the $(x, y)$ position, so the slow features (positions) are easy to be selected by a random strategy. But NOR projects the state space of the Point robot to a latent space with fast dynamics, which results in unsatisfactory performance.

|  | Point Maze | Ant Maze | Ant Push | Ant Fall | Ant Maze (Img) | Ant Push (Img) |
|---|---|---|---|---|---|---|
| LESSON | **0.05 ± 0.01** | **0.17±0.03** | **0.14±0.02** | **0.14±0.02** | **0.18±0.01** | **0.14±0.03** |
| NOR | 0.51±0.06 | 0.37±0.01 | 0.50±0.05 | 0.46±0.06 | 0.37±0.07 | 0.40±0.04 |
| Random | **0.03±0.04** | 0.94±0.11 | 0.86±0.21 | 0.90±0.08 | 0.38±0.84 | 0.46±1.00 |

Table 1: Slowness of different representations under the policy learned by LESSON, averaged over 5 randomly seeded trials with standard error. The slowness is estimated by one-step feature change $||\phi(s_t) - \phi(s_{t+1})||_2$ of 100 randomly sampled transitions.

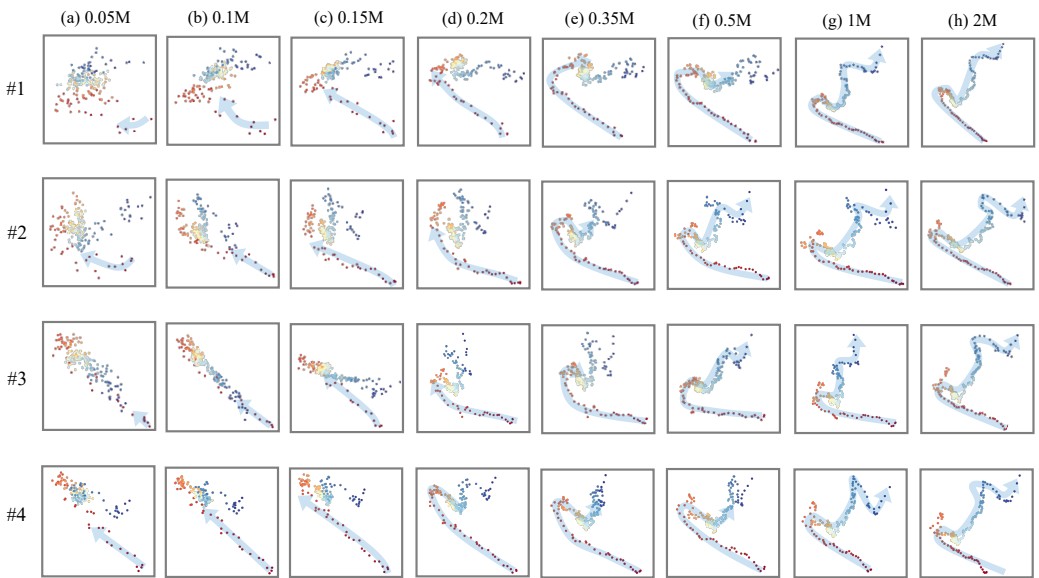

Figure 8: The representation learning process of four runs in the Ant Push (Images) task. Those visualizations demonstrate the gradually learned subgoal representations.

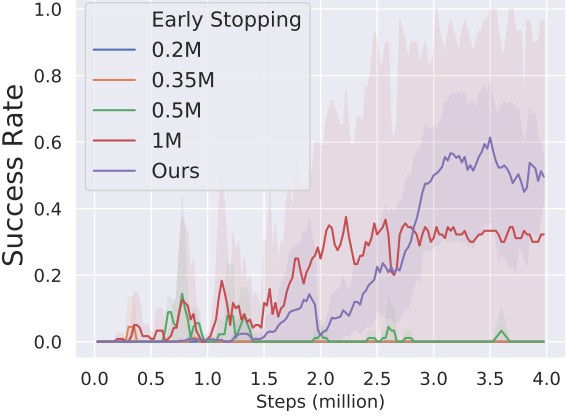

Figure 9: Early stopping the subgoal representation learning at early stages (0.2M, 0.35M, 0.5M and 1M timesteps) in the Ant Push task with visual observations. The early stopping hurts the learning performance, verifying that the subgoal representation function is gradually learned.

