# OpenReview forum: "Learning Subgoal Representations with Slow Dynamics"
_ICLR.cc/2021/Conference — ICLR 2021 Poster_

### Official Review · AnonReviewer1 · 2020-10-27
**A new objective for learning subgoal representations**

**Rating:** 7
**Confidence:** 3

**Review:**

**Summary:**
This paper proposes a new method for learning subgoal representations in HRL. The method learns a representation that emphasises features that change slowly, through a “slowness objective”. The slowness objective minimises changes in the subgoal representation between low level time steps, while maximising feature changes between the high-level temporal intervals. This objective allows for efficient exploration, which the paper justifies theoretically, and supports with some empirical experiments on challenging control domains.

**Strengths:**
The issue of subgoal selection is a critical issue for HRL, and constructing or learning a good subgoal representation on which to create subgoals is important and interrelated. Thus, any significant progress in this area is useful. The paper presents an apparently novel method that would be of interest to HRL researchers and deep RL practitioners more generally.

The paper is clearly written and the main ideas are generally very well explained (except for maybe the use of the term “state”; see weaknesses).

The paper provides some theoretical justification for the slowness objective, which is useful to support the intuition behind it. The empirical results are conducted on challenging domains and they appear strong. However, including more independent runs in each domain would strengthen the conclusions significantly (see weaknesses).

**Weaknesses:**
In sections 3.1 and 4.1 the paper uses the term "state" when describing the algorithm and other definitions. It is not clear if it is actually referring to a proper state in the Markov sense, or some approximation of a state, or even just an observation that might not be Markov at all. This is confusing, and could be clarified. How does the slowness objective interact with approximation when you do not have access to the true state (which would be most of the time)? Does the theoretical result still hold?

I think 10 runs for the NChain environment, and 5 run for Mujoco are too few. More runs would give a much better sense of the variability of the runs, greatly strengthen the conclusions about the relative performance between the algorithms. 20 runs for the NChain environment, and 10 runs for Mujoco would be a significant improvement, but even more is better.

Also, in the low run regime (~10 runs or less) sometimes it can be more informative just to plot each of the learning curves for all the runs on the same plot, along with the mean. This gives a really good sense of the variability between runs. Presenting the results this way assumes less about the distribution that the performance samples are coming from.

**Recommendation:**
Overall, I recommend to accept this paper. A clarification about the use of the term “state” and adding more runs to the experiments would increase my score.

**Questions:**
Clarify the use of the term "state" (see weaknesses).

**After Author Response and Discussion:** Thanks to the authors for their responses. After reading the other reviews and the author responses, I am raising my score to 7 (accept).

---

> ### Author Response · Authors · 2020-11-18
> **Response to Reviewer 1**
>
> Thank you for the thoughtful comments. We appreciate it if you have any further questions or comments.
>
> - Q1. The use of the term “state”.
>   - In the method section (Section 3), the term “state” can be either be a proper state in the Markov sense or an observation. Our method can generally work in both cases. In our experiments, we evaluated out method with inputs of vector states and image observations, respectively, as shown in Figure 3. Our method significantly outperforms baselines in both settings.
>   - In the theoretical analysis (Section 4), the term “state” refers to a proper state in the Markov sense. We have added a clarification in the paper. Extending the theoretical results to a more general setting will be an interesting future direction.
>
> - Q2. More runs for each task.
>
>   We have increased to 20 runs for the NChain environment and 10 runs for the Mujoco tasks. The relative performance between algorithms almost remains the same. Please see the revised version of our paper, and thanks for your suggestion. We have also added the oracle learning curves in Figure 3, as suggested by Reviewer 4.

---

### Official Review · AnonReviewer2 · 2020-10-28
**An interesting and apparently useful objective function for subgoal representation learning in hierarchical RL**

**Rating:** 6
**Confidence:** 3

**Review:**

The main idea of this paper is very nice:  that we want the features in the representation space for subgoals in HRL to be "slow" in the sense that they don't change much over primitive steps of the policy, but do change significantly over "macro" steps.  The margin-based criterion in the loss (that says we want them to change by at least m at the high level) seems well justified.

It seems like, additionally, one might want to incentivize this representation to cover the state space s well (that is, not map big chunks of s onto the same latent representation phi) and I didn't exactly see how this would be encouraged---I guess the drive for significant change at the high level accomplishes this implicitly.

Using this loss function leads to a sensible algorithm, although I was curious about why the representation learning happens at the same rate as the low-level policy learning;  I could imagine that you might want to do that at a slower time scale (more like the time scale for the high-level policy learning).

The experimental results seem to make a compelling case for the effectiveness of this algorithm.  The results for transfer learning are particularly strong---in fact transfer seems to be one of the best motivations for learning hierarchical representations, which might be harder to get to pay off in the case of learning to control a single problem instance.  Some of the explanation in this section didn't completely make sense to me:
- Why do you attribute the success of your method to improved exploration?  It seems like another very plausible explanation is that you are providing a good inductive bias for learning a policy that generalizes well (though 2 million is a lot of steps!)
- (Related) What is the purpose of the experiments on state coverage?  I don't completely see why that should be, in itself, a goal.  In fact, one attribute of good RL methods that generalize well is that they do early exploration but then focus on parts of the state space that are profitable given the particular high-level reward function they are optimizing.

All this being good, the theoretical part of the paper was quite weak.    The definitions and theorem were not rigorous and possibly incorrect.   First, from the expositional perspective, it would have helped to begin with crisp definitions and then the assumptions.  Here are many points that came up for me as I read through the text:
- What is the role of the target distribution?  Where would you get it?
- In the application of KL divergence It seems more like you'd want the p to be the target distribution.
- What is a "fact"?    Break it down into definitions and then a crisp formal assertion with a proof (if there's something in there beyond definitions).
- In Fact 1 the term "exploration process" is not defined.
- Importantly, I don't think this is well-formed unless we assume that there is no learning happening at the low level. Are we assuming that?  (If not, then the process is non-stationary).
- The high-level policy is assumed to select subgoals "randomly" :  Does this mean uniformly at random from S?   Is S bounded?
- What does it mean for the agent to be able to "move independently and identically in the state space"?
- What the X^c_t are like depends completely on the low-level policy!  It could always move left, in which case these variables would not be anything like IID.  Is this analysis supposed to be assuming that the low level is perfectly able to achieve the targets?  Or at least always successfully move c steps in the target "direction" (not sure what that means, exactly, though).
- What, precisely, do you mean by "the features are all independent"?
- Brushing assumption (a) off as a general technique to simplify analysis is not so good.  It's okay to say you're making it for now, but there's no reason to think results obtained with this assumption will generalize!
- Assumption d would require changes to the algorithm, no?
- In Theorem 1:  "r is large enough"  for what?
- I don't really understand what it means to "assume q is an isotropic Gaussian ..."  ;  that is, what exactly is q intended to be.   I see that later you say that if r is big enough it approximates a uniform, and so maybe what you're trying to show is that our exploration is uniform.  But a uniform needs to be over a bounded space.  And even with large r, q is only sort of uniform in a region bounded around the mean.
- What makes "optimal hierarchical exploration"?   I guess maybe this means:  with a fixed set of features and low-level control policy we're deciding which k features to use for subgoals, and showing that among those choices we get the best exploration (in the sense of matching a Gaussian around the origin) if we use the k slowest features.

(This is just a related thought, not really a review of this paper.  One other principle for selecting features for a subgoal representation is that they should be, in some sense, locally achievable.  Hierarchy works well when the subproblems are, in a sense, "serializable":  that we can achieve subgoal the first any way we want to, without thinking about (and/or making it harder to achieve) the next subgoal we're going to be asked to do.    One way to do this might be to encourage some "disentanglement" in the latent features, so that the policy for changing one dimension of the latent space tends not to change the other dimensions.)

In the end, I'm positively inclined toward this paper because it made me think about the concepts of what we really want in a hierarchical representation, but the mathematical exposition really needs substantial cleaning up.  But I'd rather have an interesting paper than a perfectly-executed boring one.

---

> ### Author Response · Authors · 2020-11-18
> **Response to Reviewer 2**
>
> Thank you for the inspiring comments. We appreciate it if you have any further questions or comments.
>
> - Q1. Why not learn the representation function at a slower time scale?
>
>   Our method updates the representation $\phi(s)$ per timestep with only one minibatch (100 samples), which is quite slow and steady. We compared this updating method with learning the representation at a slower time scale, e.g., updating $\phi(s)$ every $c$ timesteps with one minibatch data and updating $\phi(s)$ every $c$ timesteps with $c$ minibatches. Those methods made no significant difference. The experimental results are shown in Appendix E.
>
> - Q2. Why attribute the success of the method to improved exploration?
>   - The exploration strengths of our method are two folds. First, during the representation learning process, the generalizable representation can help the exploration of the hierarchical policy, and the samples collected by the hierarchical policy facilitate the representation learning, as shown in the newly added section (Section 6.4).
>   - Second, when the learning of the low-level policy and the representation converges, our method can achieve a larger coverage area, since the high-level policy explores in a slow subgoal space, as analyzed in Section 4. And the didactic example shown in Section 6.1 also supports this theoretical result. The larger state coverage can avoid the local sub-optimum caused by deceptive rewards.
> - Q3. The purpose of the experiments on state coverage.
>
>   The state coverage is important for exploration, especially at the early learning stage. A better state coverage can help an agent avoid stuck in a local optimum, especially when the reward function can be sparse or deceptive. Empirical results in Figure 2 show that, compared to other method, our method can quickly achieve a better state coverage at the early exploration, which allows it focus on the promising parts of the state space.
> - Q4. The theoretical part of the paper.
>
>   Thank you for the detailed comments. We have refined the theoretical part of our paper to make it more rigorous and clearer in our revision. As some of the reviewer’s questions are related to each other, we group these questions and provide corresponding clarification concisely.
>   1. Target distribution:
>
>     The target distribution is a prior distribution that we want to approximate. We assume that the target distribution is an isotropic Gaussian distribution, and as we stated in Section 4.2, when $r$ is large enough, the target distribution approximates a uniform distribution. We make this assumption since there is no prior knowledge.
> Besides, we have exchanged the notations of the target distribution and the steady distribution, and used forward KL divergence as the measure for exploration in the revised version.
>   2. Random walk:
>
>     The exploration process is defined as a process when the low-level policy is optimal, and the high-level policy chooses subgoals randomly since there is no extrinsic reward. That is why we consider this process as a random walk. We have revised Fact 1 to Definition 2 to make it clearer.
> The subgoal selection is bounded in the neighborhood of the current state. We have added a new experiment to demonstrate that our algorithm can approximately satisfy this assumption in Appendix C.
>   3. Feature independence and disentanglement:
>
>     We assume that the features are all independent so that the change in one feature dimension will not influence the others. Selecting slow features as the subgoal space can be seen as a disentanglement, in some sense. We encourage the agent to achieve subgoals with slow dynamics and ignore the fast-changing ones. It is an interesting future direction to relax this assumption. Thanks for your inspiration.
>   4. Question about Assumption (a):
>
>     We cannot release Assumption (a) in our derivation currently, but we can think deeper in future work, and we really appreciate your suggestion.

---

### Official Review · AnonReviewer3 · 2020-10-28

**Rating:** 7
**Confidence:** 2

**Review:**

This work presents a method for learning a slow-changing ("low-frequency") embedding function, which can be used as a state-abstraction function in the context of Hierarchical RL. A high-level policy, trained to solve the environments task, acts by selecting abstracted states as targets for the low-level policy which is trained as in a goal-conditioned fashion (acting in the environment itself attempting to get to the set goal).

The paper provides theoretical motivation for the focus on "slow" features in a simplified synthetic example. The method itself is evaluated on high-dim control tasks (As well as on an illustrative toy-example).

There is a lot to like about this paper. The method is elegant, building on and extending previous approaches to HRL and particularly learning subgoal representations. The use of a low-frequency (slowness) criterion for this purpose is, to the best of my knowledge, a novel contribution. The work presents strong empirical evidence for the usefulness of the approach. The paper itself is well-structured and overall well-written, easy to follow, and with good references.

I do have a few questions for which I would appreciate the authors comments:
* In principle, the loss for the embedding function (highly) depends on the behavioral policy. This is because the embedding is trained to represent "far away" states as different from each other. And, generally, the distribution of $s_t$ and $s_{t+c}$ can be rather different for, say, a random policy compared to a more purposeful/trained one. Since everything is trained jointly here (embedding is used to train the policies which in turn can change the embedding and so on), this could in principle leads to instability of the training. I think this point deserve at least some discussion.

* I find section 4 a bit confusing. "Fact 1" seems more like a definition for me, because the properties of the "random walk" itself will be *determined* by $p$ (as $p$ obviously depend on the behavioral policy. In fact from "Fact 1" we can simply say that $p$ is the steady-state/limiting/stationary distribution of the markov chain induced by the policy and the MDP). In general the relation between $p$ and the policy is not entirely clear. Note that in general, the change metric $\Delta s^i_t$ is also policy dependent. I also don't see the direct relation between $p$ in this discussion, which ultimately is an expression of the (low-level) policy, to the embedding or representation principle used throughout the paper.

* It seems the underlying assumption here is something like: the agent can independently control/change the state-features in an unconstrained way (i.e an "action" is an offset vector s.t the next state is $s_{t+1}=s_t+a_t$, but in such a way that the environment somehow 'rescales' $a_t$ so that earlier entries are smaller). If this is indeed the case, I find this section not too motivating. If there is completely no structure in the environment (state/action spaces) and everything reduced down to a uniform random-walk, we cannot learn much about even simple "realistic" RL settings.

* Having said that I think the paper presents other good reasons and motivations for the focus on slow-features. In particular I think that the example discussed in section 6.1 is important, as well as the more qualitative explanations and the relation to the unsupervised-learning literature around similar ideas.

---

> ### Author Response · Authors · 2020-11-18
> **Response to Reviewer 3**
>
> Thank you for the thoughtful and inspiring comments. We appreciate it if you have any further questions or comments.
>
> - Q1. The instability of the training.
>
>   We have added a new section (Section 6.4) to investigate the parallel learning of the representation function and the hierarchical policy. Figure 5 demonstrates that the learning of the representation function and the policy can promote each other. As the subgoal representation learning is conducted slowly and smoothly, our method is stable in most cases.
>
> - Q2. Question about Fact 1.
>
>   Thanks for your suggestion, and $p(x)$ is indeed the steady distribution of the Markov chain, induced by a random high-level policy and an optimal low-level policy. (By the way, we have exchanged the notations of the target distribution and the steady distribution in the revised version, and now $q(x)$ stands for the steady distribution.)
>
> - Q3. Is an “action” an offset vector?
>
>   No. We have made no assumption about the low-level action space, and we only consider every $c$-step change in the state space. Assuming each dimension of the change vector is bounded, we have taken the low-level action constraint into account but at a lower temporal resolution (every $c$ steps).

---

> > ### Comment · AnonReviewer3 · 2020-11-24
> > **Response to authors' comments**
> >
> > I thank the authors for their response and clarifications.
> >
> > Regarding the potential instability of training: while I definitely agree with Reviewer4 comments, mainly that the proposed explanation "*doesn't seem to be a trivial characteristic to expect from the policy and the representation*", I view the fact that this point is being explicitly discussed, together with the empirical evidence (which naturally is limited to a specific problem/s) as an important addition to the paper.
> >
> > The revised theoretical section (Sec 4) is also a bit more clear now.
> >
> > Overall after reading the author's response I would like to keep my score as before, recommending for accepting this paper.

---

> > > ### Author Response · Authors · 2020-11-24
> > > **Response to Reviewer 3**
> > >
> > > Thanks for the prompt response and additional thoughtful comments.
> > >
> > > For your reference, in response to Reviewer4’s recent comments about parallel learning, we have provided a more detailed visualization about the dynamic expansion of the explored areas along with representation learning, as shown in Figure 5 in the new revision. This figure illustrates the iterative co-adaptation of the three components, the subgoal representation improvement, the hierarchical policy learning, and the expansion of the explored area. As indicated by this result, our method does not expect that the distribution of sampled states would be good enough for training a globally proper subgoal representation, or that the representation would quickly generalize to the whole state space. Please see the latest response to Reviewer 4 for more detailed information.

---

> > > > ### Comment · AnonReviewer4 · 2020-11-24
> > > > **Little clarification/discussion**
> > > >
> > > > I agree with reviewer 3 that the parallel training has been explicitly mentioned after several reviewers have pointed it out. This should definitely be appreciated as a major addition.
> > > >
> > > > I also would like to precise regarding reviewer 3's comment that the chicken-and-egg problem is not just an "instability of training" problem. Indeed, in general, even a stable "training" may potential not espace this "learning" problem by failing to explore enough: this is actually at the heart of the exploration paradigm.
> > > >
> > > > In case this comment "our method does not expect that the distribution of sampled states would be good enough for training a globally proper subgoal representation..." is referring to my expressed concern, let me highlight that my concern was not about expecting a "globally proper subgoal representation" nor a "quickly generaliz[ing]" representation; as these should be first properly defined. However, for the sake of clarifying my comment, let's assume that "globally proper" representation means a representation that is in *some sense*\* equivalent to the xy-coordinates. In this case, I believe that such representation is being already learned (or extracted) quite early in the training (~0.15-0.2 M in Figure 12) which has to be compared to the corresponding explored area from Figure 5.
> > > > For this reason, I might find general intuitive narratives like the following "*It is reasonable that this learned representation could be generalized a little bit to the neighboring regions of the explored areas, facilitating the hierarchical policy’s exploration to further regions. As a result, the explored areas are expanded a little more.*" quite confusing and misleading.
> > > >
> > > > (*): one can thing of this equivalence of 2 representations as one being the result of local continuous distortions of the other without altering the global structure.

---

> > > > > ### Author Response · Authors · 2020-11-25
> > > > > **Response to Reviewer 4**
> > > > >
> > > > > Thank you for the additional response.
> > > > >
> > > > > - Q1. A “globally proper” representation means a representation that is in some sense equivalent to the xy-coordinates. One can think of this equivalence of 2 representations as one being the result of local continuous distortions of the other without altering the global structure. In this case, I believe that such representation is being already learned (or extracted) quite early in the training ($\sim$0.15-0.2 M in Figure 12) which has to be compared to the corresponding explored area from Figure 5.
> > > > >
> > > > >   - In fact, the “globally proper” representation defined by Reviewer 4 is **not** being already learned in the early training stages. We conducted a simple experiment to show this by early stopping the representation learning at 0.15M and 0.2M steps (corresponding to those visualizations in Figures 5 and 12), respectively. As shown in Figure 14 of Appendix F, these subgoal representations fail to learn policies to reach the goal.
> > > > >
> > > > >   - Comparing representations at $\sim$0.15-0.2M steps (subfigure c, d) with those at 2M steps (subfigure h) in Figures 5 and 12, there are dramatic changes in the global structure, not just local distortions. The yellow to blue parts of the trajectory in the representation spaces at early stages are in chaos. In contrast, the whole trajectory in the convergent representation space (Figure 5(h)) demonstrates a clearer structure of pushing the block and reaching the goal.

---

### Official Review · AnonReviewer4 · 2020-10-28
**This work would benefit from a more rigorous investigation and clearer discussion of the main claims. Interesting theoretical grounding.**

**Rating:** 4
**Confidence:** 5

**Review:**

### Summary
This paper proposes to use slow features as the subgoal representation space in the HRL setting. The proposed approach adopts a slowness objective to learn a representation (high-level action space) and use it to train goal-conditioned policies (low-level).

### Main contributions
- Slowness is an interesting property for representation learning. This work's approach to slowness is simple and easy to integrate (in practice) to the challenging setting of HRL.
- This work also makes a theoretical effort (section 4) to motivate the slowness property for subgoal space.
- The proposed method is evaluated on continuous control tasks against other references approaches to HRL.

### Main comments/concerns
- In the definition (def.1) of the measure of exploration, what justifies the choice of the reverse KL ? Is it still a good choice if q is multimodal ? Does this definition really align with your claim "a larger coverage area leads to better exploration" ?
- The algorithm implementation is not clear:
    - How is the goal sampled ? Randomly in the large domain as mentioned in the Appendix or in the neighbourhood of s as mentioned in sections 3 and 4 ?
    - What is the extrinsic reward used to train the high level policy exactly ?
- There is a chicken-and-egg problem regarding the parallel learning of the representation and its usage to train the low policies. The representation should first be trained on some area before trusting the induced training of the low policy, which requires exploring that area beforehand. It is not clear how the algorithm deals with this challenge. Moreover, the evaluation on Mujoco tasks could be misleading regarding this point (see the following comment).
- The method is evaluated on Mujoco environments that provide the xy-coordinates. The representation can learn to extract the coordinates by only training on a limited area (e.g. neighbourhood of initial state) -- being able to early stop the representation training seems to support this hypothesis. Was this phenomenon studied is your experiments ? How is it limiting the proposed method beyond this type of environments ? How does the proposed method work when this inductive bias can't be leveraged, and when exploring an area of the state space is required to build a useful representation of it ?
- Figure 4 shows the learned representation. Was it learned from the states or the images setting ? If learned from states, how does it look like when learned from images ? And, related to the previous comment, how does it look like along the training ?
- How was the oracle trained ? It would be interesting to see the oracle training curves along with the compared methods on Figure 3.
- How reliable/relevant is the slowness evaluation (section 6.3) in comparing the different methods ? It seems that the slowness objective can learn arbitrarily slow features (according to your measure) by weighting the attraction term more than the repulsive one (tuning m and c can also influence the slowness measure).
- "As the state space of the Point robot is simple, the dynamics of the randomly selected features are slow as well": What do you mean by simple state space ?
- "the high-level policy guides the agent to jump out of the local optimum of moving towards the goal" : local optimum w.r.t which reward function ?
- "The transfer effect [...] is more significant [...] since the target task, Ant FourRooms, is of a larger size and more difficult": Is the transfer effect confirmed when transferring from AntPush and AntFall to the larger Ant FourRooms ?

### Minor comments
- The AntPush performance curves are limited to 2.5 and 2 million steps. How was this horizon decided for this environment ? How do these curves look like for a longer training (4 million steps) ?
-  Equation 7 seems confusing: p(x) is the asymptotic distribution of Y_n, meaning that it should not depend on n as shown in equation 7.
- Why does Figure 5(a) has a 4 millions step while 5(b) has 8 million steps ?

Clarity: The paper reads well, but the experiments and algorithm presentation is sometimes unclear if not confusing.

---

> ### Author Response · Authors · 2020-11-18
> **Response to Reviewer 4 (Part 1)**
>
> Thank you for the detailed comments. We have refined our paper to make it more rigorous and clearer. We have also added some new experimental results to clarify the reviewer’s questions. We appreciate it if you have any further questions or comments.
>
> - Q1. The measure of exploration in def. 1.
>
>   Thank you for the suggestion. Our theoretical claim also holds with the forward KL definition, as discussed in the revision. We used reverse KL to measure coverage, as we assume the target state distribution is Gaussian in our analysis, which is unimodal. We agree with the reviewer that, for a multimodal target state distribution, using forward KL to measure exploration may be more proper. We have added discussions in our revision.
>
> - Q2. Is the goal sampled in a large domain or in the neighborhood of s?
>
>   We have tested these two methods for sampling subgoals. They demonstrated similar performance, as shown in Appendix C. This is because distant subgoals are hard to reach for the low-level policy, the high-level agent will learn to selects subgoals nearby the current latent state during the learning process. For empirical results in our paper, we used the way described in the Appendix, and have corrected the description in sections 3 and 4 in the revision. We have clarified this point in Appendix B.3.
>
> - Q3. What is the extrinsic reward?
>
>   We used a common reward setting (e.g. [1,2,3]), where the extrinsic reward is given as the negative L2 distance to the environment goal. This extrinsic reward is deceptive, which can lead the learning to a local sub-optimum. The hierarchical structure can enable better exploration with larger state coverage and avoid such local optima.
>
> - Q4. The chicken-and-egg problem regarding the parallel learning of the representation and low-level policies.
>
>   Empirical results show that the iterative learning of the subgoal representation and the low-level policies promotes each other. This is because a low-level policy, even not very good, will collect useful data for learning a good subgoal representation, which is updated in a slow and stable manner and will also generalize to other parts of the state space. This gradually improved representation provides dense rewards to the low level, and the trajectories collected by the learned policy are utilized to train the representation. This co-adaptation process is demonstrated in Figure 5 (a)~(c) in the revision.
>
> - Q5. How does the proposed method perform beyond environments providing the xy coordinates?
>   - Our approach also works with visual observations **without** coordinate information, since the proposed inductive bias (slow dynamics) has a good generalization ability.
>   - The training process of the representation learned from images **without** coordinates are shown in Section 6.4 in the revision. When the hierarchical agent has learned how to reach the easy goal near the initial states, the trajectory embeddings in this area (red to yellow part in Figure 5 (a)$\sim$(c)) align with the coordinates. And the learned representation can be generalized to further areas, although a little chaotic, shown by the yellow to blue trajectory part in Figure 5 (d)$\sim$(f). This generalizable subgoal representation can facilitate the hierarchical policy exploration for the distant hard goal. By the way, early stopping the representation training is to make the learning faster in terms of wall time. We have added an ablation study of early stopping in Appendix D.
>
> - Q6. What was the representation learned from in Figure 4?
>
>   It was learned from the states. We have added the representation learned from images along with the training in Section 6.4. As we stated in Q4, the representation learned from images can facilitate policy learning, and vice versa.
>
>
> [1] Nachum et al. “Near-Optimal Representation Learning for Hierarchical Reinforcement Learning.” In ICLR, 2019.
>
> [2] Nachum et al. “Data-Efficient Hierarchical Reinforcement Learning.” In NeurIPS 2018.
>
> [3] Jinnai et al. “Exploration in Reinforcement Learning with Deep Covering Options.” In ICLR, 2020.

---

> > ### Author Response · Authors · 2020-11-18
> > **Response to Reviewer 4 (Part 2)**
> >
> > - Q7. How was the oracle trained? Show and compare to the oracle training curves.
> >   - The oracle was trained with the xy-coordinates as the subgoal space. We have added the oracle training curves to Figure 3 in the revision, which are averaged by 10 runs (as suggested by Reviewer 1). The dashed lines in the previous version are the average top ten success rates for the oracle method.
> >   - In the traditional MuJoCo tasks (where xy-coordinates are included in the state representation), our method can efficiently learn a proper subgoal representation and has a similar performance with the oracle curves. In the image-input tasks without xy-coordinates, as expected, the learning curve of our method is a slight slower than the oracle method, but it can achieve a similar final performance as the oracle method.
> >
> > - Q8. The role of the slowness evaluation in Section 6.3.
> >
> >   The slowness evaluation used in Section 6.3 is to explore the possible connection between the performance and the slowness of learned subgoal features of different methods, e.g., whether a subgoal representation learning method that has a better performance on a task learns slower features. Such evaluation can provide a potential motivation for using a slowness objective for the subgoal representation learning. We would like to emphasize that this evaluation is not saying that a method learning slower features is better, which does not make sense, because we can always find a method to learn arbitrarily slow features (as the reviewer also pointed out).
> >
> >
> > - Q9. The meaning of “simple” state space of the Point robot.
> >
> >   The state space of the Point robot is relatively low-dimensional and contains little information other than the xy-coordinates, so the slow features (xy) are easy to be selected by a random strategy.
> >
> > - Q10. "the high-level policy guides the agent to jump out of the local optimum of moving towards the goal": local optimum w.r.t which reward function?
> >
> >   The local optimum is w.r.t the extrinsic reward function for the high-level, i.e., the negative L2 distance to the environment goal, as we stated in Q3.
> >
> > - Q11. The transfer effect from Ant Push to Ant FourRooms.
> >
> >   We have added the transfer experiment from Ant Push to Ant FourRooms in Figure 6, which also achieved a more efficient learning process than from scratch. We have refined the discussion of the transfer results in Section 6.5.
> >
> > - Q12. The varying training steps in different tasks.
> >
> >   We have increased the learning horizon to 4 million steps in the Ant Push tasks in Figure 3. Similarly, we have plotted all the transfer learning curves with an 8 million step horizon in the revised version. We plot different tasks with different training horizons because some tasks are relatively simpler and converge faster than other tasks.
> >
> > - Q13. A confusing point: the asymptotic distribution of $Y_n$.
> >
> >   Thank you for pointing out this typo, and we have fixed it.

---

> > > ### Comment · AnonReviewer4 · 2020-11-23
> > > **The parallel iterative learning of representation and policy**
> > >
> > > Thanks for your answers and clarifications. Here are some additional comments on the main remaining concern.
> > >
> > > - Regarding the chicken-and-egg problem your answer was:
> > > "*This is because a low-level policy, even not very good, will collect useful data for learning a good subgoal representation, which is updated in a slow and stable manner and will also generalize to other parts of the state space.*"
> > >   - It seems that the main explanation stands on the hope that the distribution of the sampled states would be "useful" and, more importantly, the fact that the representation would generalize to other parts of the state space. This doesn't seem to be a trivial characteristic to expect from the policy and the representation. Also, while claiming improving exploration, the paper doesn't do justice to this exploration problem (data sampling for the representation and the policy) by e.g. investigating/validating their understanding of it more explicitly.
> > >    - Also, the results with the visual observation show a partial success (~ 0.4-0.5). How do you explain this performance ? Did the representation or the policy not converge to "good" ones ? Can you precise if figure 5 correspond to a successful run or is it what we would consistently observe by running the algorithm ? Moreover, did you try higher resolutions than 5x5 ? Does the proposed method suffer less (benefit more ?) from coarser resolutions than the other algorithms ?

---

> > > > ### Author Response · Authors · 2020-11-24
> > > > **Response to the question about the parallel learning**
> > > >
> > > > Thank you for the prompt response and additional thoughtful comments.
> > > >
> > > > - Q1. It seems that the main explanation stands on the hope that the distribution of the sampled states would be "useful" and, more importantly, the fact that the representation would generalize to other parts of the state space.  This doesn't seem to be a trivial characteristic to expect from the policy and the representation.
> > > >
> > > >   The word "useful" might be over-interpreted. Our method does not expect that the distribution of sampled states would be good enough for training a globally proper subgoal representation, or that the representation would quickly generalize to the whole state space. Instead, at the beginning of the learning, the states collected by a nearly random exploration policy are enough to train an inaccurate but useful subgoal representation for the area around these sampled states, as shown in Figure 5(a). It is reasonable that this learned representation could be generalized a little bit to the neighboring regions of the explored areas, facilitating the hierarchical policy’s exploration to further regions. As a result, the explored areas are expanded a little more. Then the newly sampled states in the expanded region are utilized to improve the subgoal representation as well. Figure 5 of our new revision illustrates this iterative co-adaptation of the three components, the subgoal representation improvement, the hierarchical policy learning, and the expansion of the explored area.
> > > >
> > > > - Q2. The results with the visual observation show a partial success ($\sim$ 0.4-0.5). How do you explain this performance?
> > > >
> > > >   Since this environment is a partially observable MDP, the convergent success rate can hardly reach 1. The performance of our method approaches that of the oracle case at nearly 4 million timesteps, when they achieve a success rate $\sim$ 0.4-0.5, as shown in Figure 3. With a longer training horizon, our method can converge to a success rate of about 0.7 $\sim$ 0.8. Please refer to Figure 11 in Appendix F.
> > > >
> > > > - Q3. Are the experimental results in Figure 5 consistently observable by running the algorithm?
> > > >
> > > >   Yes. We have visualized the representation learning of four runs in the Ant Push (Images) task in Figure 12 of Appendix F. Those visualizations demonstrate the gradually learned subgoal representations.
> > > >
> > > > - Q4. Did you try higher resolutions than 5x5?
> > > >
> > > >   Yes. In Ant Maze (Images) tasks with different resolutions (4x4, 5x5, 6x6, 8x8), our method consistently outperforms the baseline methods. In the tasks with higher resolutions, the performance of our method, NOR, and Oracle have an improvement. Those experimental results are shown in Figure 13 of Appendix F.

---

> > > > > ### Comment · AnonReviewer4 · 2020-11-24
> > > > > **Thanks for the clarification**
> > > > >
> > > > > Thank you for the additional experimental results. I do appreciate that the authors took several suggestions into account for their more recent revisions.
> > > > >
> > > > > I'd like to also thank them for the clarifications brought in this discussion. These will be useful for the following reviewing phase.

---

### Comment · Area_Chair1 · 2020-11-19
**Better positioning the paper in the literature**

Dear authors,

I noticed that some apparently relevant literature is not being cited in the current version of the manuscript. Kompella et al. (2017), for example, introduced the idea of using slow feature analysis for skill acquisition. Sprekeler (2011) has shown the equivalence, given some conditions, between slow feature analysis and laplacian eigenmaps, which are related to proto-value functions (Mahadevan and Maggioni 2007) and the successor representation (Dayan 1993), which are all methods that have been used in the past to also discover skills (Machado et al. 2017, 2018; Bar et al. 2020). Although I realize several of these methods wouldn't be easily adapted to the problems tackled in this paper, it might be useful to better position the paper to give readers a big picture view of the field, particularly Kompella et al.'s work. Am I missing something that makes these approaches not related? I'm not suggesting that you should survey the whole field, but I'm trying to understand why this part of the literature was ignored.

* Amitay Bar, Ronen Talmon, Ron Meir: Option Discovery in the Absence of Rewards with Manifold Analysis. ICML 2020
* Peter Dayan: Improving Generalization for Temporal Difference Learning: The Successor Representation. Neural Comput. 5(4): 613-624 (1993)
* Varun Raj Kompella, Marijn F. Stollenga, Matthew D. Luciw, Jürgen Schmidhuber: Continual curiosity-driven skill acquisition from high-dimensional video inputs for humanoid robots. Artif. Intell. 247: 313-335 (2017)
* Marlos C. Machado, Marc G. Bellemare, Michael H. Bowling: A Laplacian Framework for Option Discovery in Reinforcement Learning. ICML 2017: 2295-2304
* Marlos C. Machado, Clemens Rosenbaum, Xiaoxiao Guo, Miao Liu, Gerald Tesauro, Murray Campbell: Eigenoption Discovery through the Deep Successor Representation. ICLR 2018
*Sridhar Mahadevan, Mauro Maggioni: Proto-value Functions: A Laplacian Framework for Learning Representation and Control in Markov Decision Processes. J. Mach. Learn. Res. 8: 2169-2231 (2007)
*Rahul Ramesh, Manan Tomar, Balaraman Ravindran: Successor Options: An Option Discovery Framework for Reinforcement Learning. IJCAI 2019: 3304-3310
*Henning Sprekeler: On the Relation of Slow Feature Analysis and Laplacian Eigenmaps. Neural Comput. 23(12): 3287-3302 (2011)

---

> ### Author Response · Authors · 2020-11-20
> **More Related Work Added**
>
> Dear Area Chair,
>
> Thank you for pointing out the missing relevant literature. We have added a paragraph in Section 5 to discuss the skill discovery approaches using slow feature analysis. We have also included some other related work about eigenoptions [1, 2, 3] and deep successor representation [4].
>
> Our method and those skill discovery methods share some similarities in learning low-level policies in a smooth/slow latent space. However, the skill learning methods can be regarded as bottom-up HRL, where a set of task-agnostic low-level skills are firstly learned with some intrinsic reward functions and then composed to solve downstream tasks. In contrast, our goal-conditioned method can be regarded as top-down HRL, where the high-level policy sets subgoals to the low level during learning a task, and the level-level policy is incentivized to reach those subgoals.
>
> It would be an interesting future research direction to explore bottleneck or termination state detection using the graph Laplacian [2] in goal-conditioned HRL, since those bottleneck states are critical to efficient exploration. We appreciate it if you have any further questions or comments.
>
>
> [1] Jinnai et al. “Discovering Options for Exploration by Minimizing Cover Time.” in ICML 2019
>
> [2] Wu et al. “The Laplacian in RL: Learning Representations with Efficient Approximations.” in ICLR 2019
>
> [3] Jinnai et al. “Exploration in Reinforcement Learning with Deep Covering Options.” in ICLR 2020
>
> [4] Kulkarni et al. “Deep Successor Reinforcement Learning” arxiv 2016.

---

### Author Response · Authors · 2020-11-22
**Summary of the Revision**

Thank you to all the reviewers for insightful comments, which helped us improve our work. Here is a brief summary of major updates made to the revision:

1. Added visualization of the parallel learning process of the subgoal representation and the hierarchical policy with image observations.

2. Refined our theoretical claim from reverse KL to forward KL.

3. Added the number of runs in the NChain task to 20 and the Mujoco tasks to 10.

4. Added additional discussions in related work about skill discovery using slow feature analysis.

5. Added more ablation studies on the high-level action space, early stopping, and different updating frequency of the representation function.

6. Added some clarifications to the theoretical analysis section to make it more rigorous.

We are happy to provide further clarification if you have any additional questions or comments.

---

### Decision · Program_Chairs · 2021-01-07
**Final Decision**

**Decision:**

Accept (Poster)

**Comment:**

This paper introduces an HRL method that uses slow features to define subgoals (or abstract states), which can then be used by goal-conditioned policies. It is said that such an approach allows for efficient exploration. Most reviewers are recommending the acceptance of this paper, they found the method interesting and they think it introduces interesting ideas that are not that common to the HRL literature. Thus, I’m recommending the acceptance of this paper.

I’d still encourage the authors to take the reviewers comments into consideration when preparing the final version of the paper. Specifically, it would be useful to explicitly discuss the “chicken and egg problem” and the fact that the agent has access to a function defining the distance to the goal before the goal was observed for the first time. Some baselines have the same assumption, but it is somewhat weird to discuss exploration in this setting without further clarifications.